# Study of thin layer film evolution near bacterial cellulose membrane by Ag|AgCl electrodes in chamber with lower concentration

Sławomir Grzegorczyn[1]*, Andrzej Ślęzak[2]

**1** Department of Biophysics, School of Medicine with the Division of Dentistry in Zabrze, Medical University of Silesia, Zabrze, Poland, **2** Department of Health Science, Jan Dlugosz University, Częstochowa, Poland

* grzegorczyn@sum.edu.pl

**Data Availability Statement:** All relevant data are within the manuscript and its Supporting Information files.

## Abstract

We used the method of measuring potential difference between two Ag|AgCl electrodes immersed directly into electrolyte solution with lower concentration and at different distances from membrane. The bacterial cellulose membrane was placed in horizontal plane in the membrane system with configurations with higher NaCl concentration and density under (A) and over the membrane (B). In both configurations at the initial moment the voltage between electrodes amounted to zero. After turning off mechanical stirring of solutions, in configuration A we observed the monotonic increase and next stabilization of voltage while in configuration B after short time dependent on the initial quotient of NaCl concentrations on the membrane we observed appearance of pulsations of measured voltage and gradual decrease of mean voltage over time. Smooth changes of voltage are connected with diffusional reconstruction of Concentration Boundary Layers (CBLs) while fast increase and subsequent pulsations of voltage are connected with the appearance of hydrodynamic instabilities (gravitational convection) near membrane imposed on diffusive reconstruction of thin layer. The time needed for the appearance of hydrodynamic instabilities in CBL depended nonlinearly on the initial ratio of electrolyte concentrations on the membrane.

## 1. Introduction

A membrane separates the volumes of a system and allows for controlling an exchange of solvent and dissolved solutes. The importance of membranes in biology is undeniable and proper functioning of complex membranes determines the physiological states of cells, organs and tissues. Membranes are also used in different technologies to control distributions of substances in chambers of technical systems. Application of membranes in technical processes such as ultrafiltration and nanofiltration [1] osmosis, reverse osmosis [2], power generation [3], makes it possible to achieve fresh water, pure selected substances and generate electrical power. These processes pollute the environment to a small extent. For this reason the membrane processes are important today and will be all the more important in the future. One of the problems with

**Funding:** Sławomir Grzegorczyn KNW-1-044/N/8/I This work was supported by Medical University of Silesia No.

**Competing interests:** The authors have declared that no competing interests exist.

membrane transport is fouling of membrane. This process is connected with gradients of solutes concentrations appearing at membrane surfaces. The layers near membrane surfaces are called Concentration Boundary Layers (CBLs) [4, 5]. In most cases, this is disadvantageous phenomenon causing decrease of thermodynamic forces on the membrane and fluxes of solutes through the membrane. In all membrane processes the diffusion is responsible for the reconstruction of CBLs and leads to the increase of thicknesses of CBLs. In technology many different solutions to this problem are used, which leads to decrease of CBLs influence on the membrane transport [6]. One of the interesting phenomena leading to the decrease of CBLs thicknesses is the gravitational convection [7]. This phenomenon is connected with appearance of sufficiently large gradients of solutions densities in CBLs directed opposite to the gravitational acceleration vector. Gravitational convection also called hydrodynamic instabilities in near membrane areas were studied by means of observation of change over time of electrical voltage between electrodes immersed in solutions [8] or by means of optical methods, such as Mach-Zender interferometry [9]. The way of reconstruction of CBLs depends on the membrane structure, for example its porosity, bounded charges with structure of the membrane and tortuosity of membrane channels [10]. One of the membranes with electrically neutral structure and suitably high porosity making it possible to study convection phenomenon is the bacterial cellulose membrane. This membrane is frequently used in medicine as wound cover because of its high hydrophilic properties, good adhesion to skin and efficient transport of water and small molecules [11–13]. Besides, this membrane as a wound cover is an impenetrable barrier for bacteria, which is why under this cover the appropriate wound healing environment is stabilized, especially for wounds difficult to heal, such as burns and ulcers [14, 15]. The range of medical applications of bacterial cellulose materials is constantly expanding and the most important applications of BC include drug delivery, vascular grafts, scaffolds for tissue engineering [16]. In addition, bacterial cellulose dressings show great water holding capability, have transparent nature and good mechanical properties [17].

Taking into consideration the properties of bacterial cellulose membrane we used the voltage measurement methods in the membrane system with bacterial cellulose membrane located in horizontal plane. In this case the influence of gravity on the appearance of hydrodynamic instabilities in CBLs is maximal. In our study we extend the method of voltage measurement in the membrane systems [18–20] to the case with both electrodes in one chamber with lower ions concentration. The electrode located at the membrane surface is the working electrode while the electrode distant from the membrane is the reference electrode. The advantage of this method is the dependence of changes of the observed voltage only on concentration changes in CBLs in one chamber compared to the case of electrodes on both sides of the membrane, where the potential drop on the membrane must also be taken into account [8]. As in the previous studies we take into consideration two configurations of the membrane system, with higher density under (A) and over (B) the membrane. Time characteristics of voltage changes in the membrane system with different initial gradients of solute concentration on the membrane are the starting points when we calculate concentration characteristics of voltages in the steady states. Using the Kedem-Katchalsky formalism of the membrane transport we applied the previously developed model for inhomogeneous solutions [18, 20] to calculate changes of concentration distribution in the chamber with lower concentration. Besides, for B configuration of the membrane system the time needed for reconstruction of suitably high gradients of densities in CBLs to appear hydrodynamic instabilities was measured and calculated from the model and the concentration Rayleigh number. This allows us to characterize the conditions of appearance of gravitational convection in the membrane systems.

## 2. Theory

Modelling of CBLs evolution near the membrane is connected with the description of solutes transport through the membrane and the diffusion of solutes in solutions at the membrane surfaces. One way to describe such processes is the connection of membrane transport described by Kedem-Katchalsky equations with the diffusion equation for solutes transport in CBLs [18]. Such a method allows for description of heterogeneous conditions of membrane transport with the appearance of CBLs in the membrane system. The Kedem-Katchalsky equations for the case of mechanical pressure difference through the membrane equal to zero ($\Delta P = 0$) can be presented in the form [21]

$$J_v = L_p\left(-\sum_{j=1}^{n}\sigma_j \cdot \Delta\pi_j + \beta \cdot I\right) \tag{1}$$

$$J_s = \bar{C}_s(1 - \sigma_s)J_v + \sum_{j=1}^{n}\omega_{sj}\Delta\pi_j + \frac{t_s}{z_s F}\cdot I \tag{2}$$

$$I = -\kappa \cdot \beta \cdot J_v + \kappa \cdot \sum_{j=1}^{n}\frac{t_j}{z_j F}\frac{RT}{C_j}\Delta C_j + \kappa \cdot E \tag{3}$$

where $J_v$ and $J_s$ are the volume and ion fluxes ($s$–indexes for suitable ions, $n$ number of types of ions in solution), $I$ is the density current through the membrane, $\Delta\pi_j$ is the osmotic pressure difference through the membrane for $j$ solute, $\bar{C}_s = (C_h - C_l)[\ln(C_h C_l^{-1})]^{-1}$ is an average $s$ ions concentrations in the membrane and $E = \frac{\Delta\bar{\mu}}{z_s F}$ is the gradient of electrical potential on the membrane. Besides, $C_h$ and $C_l$ ($C_h > C_l$) are the concentrations of ions in chambers at the initial moment, $L_p$, $\sigma_s$ and $\omega_s$ are hydraulic, reflection and solute permeability coefficients for membrane respectively. $\beta$, $t_s$ and $\kappa$ are electroosmotic coefficients, transference number of ions $s$ and conductivity of the membrane. Besides, $F$, $R$ and $T$ are Faraday number, gas constant and absolute temperature respectively, and $\bar{\mu}$ is the electrochemical potential of the solution.

Diffusion of solute through CBLs can be described by an equation

$$\frac{\partial C}{\partial t} = -\frac{\partial J_s}{\partial x} = D_s \cdot \frac{\partial^2 C}{\partial x^2} \tag{4}$$

where $D_s$ is the diffusion coefficient of solute ($s$) in aqueous solution.

Taking into consideration Eqs (1)–(4) and assumption that the current through the membrane during measurement of voltage between electrodes is equal to zero the differential form of Eqs (1)–(4) were elaborated [18, 20]. During passive measurement of voltage by millivoltmeter with high input impedance the current between electrodes is negligible. We also assumed that the surface of the membrane is equal to the cross-sectional area of chambers containing solutions. Taking the above assumptions and Eqs (1)–(4) the difference equations for layers in chambers can be written as [20]

$$C_{i,1}^{k+1} = C_{i,1}^{k} + (-1)^{i+1}\frac{\Delta t}{d_w}\cdot\left(B^k RT(C_{i,1}^{k} - C_{i,1}^{k})\right) - \frac{\Delta t}{d_w^2}D_s(C_{i,1}^{k} - C_{i,2}^{k}) \tag{5}$$

$$C_{i,n}^{k+1} = C_{i,n}^{k} + \frac{\Delta t}{d_w^2}D_s\left(C_{i,n+1}^{k} + C_{i,n-1}^{k} - 2C_{i,n}^{k}\right) \tag{6}$$

where $C_{i,1}^{k+1}$ are the concentrations in layers adjacent to the membrane (second subscript $n = 1$) suitably in chamber with lower concentration ($i = 1$) and in chamber with higher

concentration ($i = 2$) at point of time $k$, $B^k = \omega_s - \sigma_s L_p (1 - \sigma_s) \cdot \bar{C}^k$ and $\bar{C}^k = (C_{2,1}^k - C_{1,1}^k) \cdot [ln(C_{2,1}^k \cdot (C_{1,1}^k)^{-1})]^{-1}$ is the average concentration in the membrane. Moreover, $d_w$ is the thickness of the layer and $\Delta t$ is a time interval used in the recursive method of calculation. Numerical solutions of differential Eqs (5)–(6) allow us to determine changes of concentrations in time at any point in CBLs. The parameter characterizing CBL is its thickness. In the model presented above the CBLs thickness ($\delta$) was determined on the basis of the Lerche criterion [22]

$$\frac{C_o - C_\delta}{C_o - C_m} = K \tag{7}$$

where $C_o$, $C_m$ and $C_\delta$ are the solute concentrations in the chamber, at the membrane surface and at the distance $\delta$ from the membrane respectively. In our calculations we assumed $K$ equal to 0.01 [22, 23]. The above-mentioned model allows us to describe the diffusional conditions of CBLs evolution in time but in some cases (B configurations of the membrane system) in CBLs there can appear gradients of densities directed opposite to the gravitational acceleration. These cases can be observed when the density of solution depends on the solute concentration. For aqueous NaCl solutions an increase in NaCl concentration causes an increase of solution density. For suitably high gradients of density the macroscopic movement of solutions, so-called the gravitational convection (or hydrodynamic instability) in the membrane system can be observed [19, 24, 25]. These processes blur CBLs, decreasing concentration and density gradients in CBLs and thicknesses of CBLs. In order to describe the moment of appearance of hydrodynamic instability in the membrane system we used the concentration Rayleigh number for CBL, which can be written as [26, 27]

$$Ra = \frac{g \frac{\partial \rho}{\partial C} \Delta C \delta^3}{\rho v D_s} \tag{8}$$

where $g$ is the gravitational acceleration, $\Delta C$ is the difference of concentrations in CBL, $\delta$ is the thickness of CBL, $\rho$ is the density of solution, $v$ is the kinematic viscosity coefficient for solution and $D_s$ is the diffusion coefficient for the solute in solution. The identification of the moment of appearance of the hydrodynamic instability in CBLs can be done by the moment when the concentration Rayleigh number achieves its critical value [28]. The critical value of concentration Rayleigh number depends on the conditions in which the gravitational convection appears. Conditions of gravitational convection in the analyzed membrane system are of the type: rigid surface (membrane surface) and free surface (border of CBL in the chamber). In this case, the critical value of the Rayleigh number for CBL can be assumed as $(Ra)_c = 1100.6$ [29]. Time needed for appearance of hydrodynamic convective instabilities ($t = T_p$) can be determined when the Rayleigh number for CBL reaches its critical value $(Ra)_c$ [19].

One of the experimental methods used in order to monitor the evolution of CBLs is the measurement of voltage between electrodes directly immersed into solutions. Hitherto existing studies of measurement of voltage in the membrane system assumed location of electrodes on both sides of the membrane. In this case the measured voltage includes the electrical potential difference on the membrane as a constituent part. In the case analyzed in this article we located the electrodes in the same chamber of the membrane system, one of the Ag|AgCl electrodes was near the membrane, 5mm from the membrane surface (working electrode) while the other was located 35mm from the membrane surface (reference electrode). The voltage measured between the Ag|AgCl electrodes located in solution with a lower NaCl concentration and in different distances from the membrane surface can be derived from the definition of

difference of potentials between points in solutions [30] and written in the following form

$$\Delta\psi^k = -\frac{2RT}{F}\left[t_+ \cdot ln\left(\frac{\gamma_{l,50}^k \cdot C_{l,50}^k}{\gamma_{l,350}^k \cdot C_{l,350}^k}\right)\right] \tag{9}$$

where $\gamma_{l,n}^k \cdot C_{l,n}^k$ ($l$–is for the chamber with lower concentration) are the products of ion activity coefficients and concentrations at electrodes surfaces, located in the chambers with lower concentration: 5 mm ($n = 50$) and 35 mm ($n = 350$) from the membrane surface. Besides, $t_+$ is the apparent transference number for Na$^+$ ions in the solution, while $R$, $T$, $F$ are the gas constant, thermodynamic temperature and the Faraday constant respectively. Solutions in the chambers are homogeneous during mechanical stirring, so at the initial moment of measurements we can assume that for all $n$: $C_{l,n}^0 = C_l$, $C_{h,n}^0 = C_h$ (initial conditions). Turning off mechanical stirring of solutions is the initial moment of the measurement of voltage between electrodes in the membrane system and the above conditions are the initial conditions for calculation of the distribution of concentrations from the model based on Eqs (5)–(6). After turning off mechanical stirring, the solutions near the membrane became non-homogeneous with CBLs at the membrane surfaces growing over time. In a steady state of the membrane system, reached after suitably long time, this phenomenon, called concentration polarization of the membrane, makes the concentrations on both sides of the membrane almost the same.

Taking into consideration the localization of electrodes in one of the chambers we can notice that in agreement with the definition of potential difference between electrodes in solution [30] the voltage caused by difference of concentrations at electrode surfaces can be written as

$$d(\psi) = -\frac{2RT}{F}\left[t_+ \cdot d[ln(a)]\right] = -\frac{2RT}{F}t_+ \cdot \frac{da}{a} \tag{10}$$

where $a$ is the activity of solute ($a = \gamma C$). When we treat $da$ as a difference of ion activities at electrode surfaces we can notice that for greater activity ($a$) of ions in solution in the chamber where electrodes are located the lower values of the observed potential ($d\psi$) are measured. This is why electrodes should be always located in the chamber with lower concentration. For example in the membrane system in chambers with NaCl concentrations equal to 0.01 mol/m$^3$ and 10 mol/m$^3$ respectively, the observed maximal voltages for electrodes in the chamber with lower concentration (0.01 mol/m$^3$) were about tens of mV while for electrodes in the chamber with higher concentration (10 mol/m$^3$) a few mV.

## 3. Materials and methods

The membrane system consisted of two cylindrical chambers with volumes V = 200 cm$^3$ each and bacterial cellulose membrane with a surface S = 6.1 cm$^2$, separating the chambers and located in the horizontal plane. The Bacterial Cellulose membrane (*Biofill*, Fibrocel Productos Biotechnologicos Ltd. Curitiba, Brazil) was used. Bacterial cellulose was produced by the bacterial strain *Acetobacter Xylinum*, by a static method. After bacterial cellulose had been purified, membrane sheets were obtained by compressing multiple layers of bacterial cellulose under high pressure with draining the water. The phenomenological coefficients of the bacterial cellulose membrane measured in accordance with procedures described in [5] amount to: hydraulic permeability coefficient $L_p = 6.4 \times 10^{-11}$ m$^3$ N$^{-1}$ s$^{-1}$, reflection coefficient $\sigma_s = 0.0034$ and coefficient of NaCl diffusion permeability $\omega_s = 17.1 \times 10^{-10}$ mol N$^{-1}$ s$^{-1}$. The structure of the membrane (Fig 1B) is in the form of a multilayer networks of interwoven cellulose microfibers, compressed under sufficiently high pressure and with drained water. Cellulose fibers have a cross-sectional diameter of 0.1 to 0.2 μm, while their length is several μm. This structure

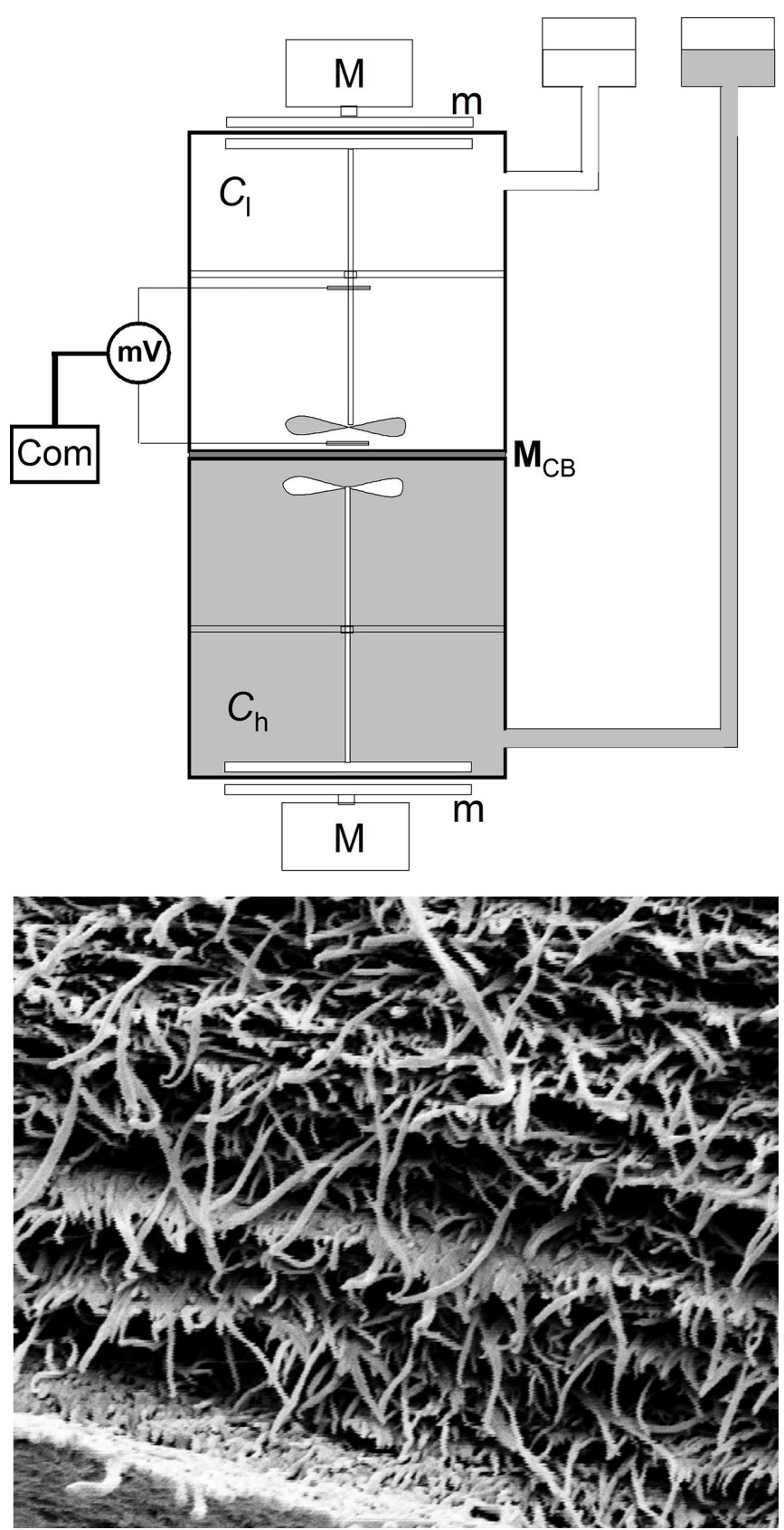

**Fig 1.** A–measurement system in configuration A; $M_{CB}$–bacterial cellulose membrane, M - motor, m–magnet, mV–milivoltmeter, Com–computer. B—Cross-section of dry bacterial cellulose membrane (*Biofill*) and part of the surface of the membrane, obtained by means of a Zeiss Supra 35 with magnification 23 450.

of the network of intertwined fibers gives the membrane great flexibility and tear strength. The membrane has a layered structure, each layer consisting of cellulose fibers arranged in different directions relative to each other. The thickness of the membrane in dry state can be estimated at about 10 μm [5], while the hydration of the membrane causes its thickness to almost double to 18–20 μm [7]. The multilayer structure of the membrane with entangled fibers of bacterial cellulose makes the holes through which the solution penetrates small, of the order of 0.1 μm, with high tortuosity along the entire length of the membrane. For this reason, this membrane, as a dressing, is an excellent barrier to bacteria [11].

Coefficient of NaCl diffusion in aqueous solutions amounts to $D_s = 14.7 \times 10^{-10}$ m$^2$ s$^{-1}$. The simplified scheme of the measurement system for both of the analyzed configurations is presented in Fig 1A. Other elements of the measurement system (e.g. solution mixing system) were described in [31].

Aqueous NaCl solutions of different concentrations were used, but in the chamber with a lower NaCl concentration there was always solution with NaCl concentration at the initial moment: $C_l = 10^{-5}$ mol/l. On the other hand, the NaCl concentrations at the initial moment in the second chamber ($C_h$) were established in the range: from $10^{-4}$ mol/l to $5 \cdot 10^{-2}$ mol/l. The relative error of solution preparation was lower than 1%. Two configurations of the membrane system were distinguished: configuration A, in which at the initial moment the NaCl solution with a lower concentration (and lower density) was above the membrane and configuration B, in which at the initial moment, the NaCl solution with a higher concentration was above the membrane. To monitor concentration changes in the membrane system, we used a method of measuring the voltage between Ag|AgCl electrodes immersed directly in the solution in the chamber with a lower concentration. One of the electrodes (active electrode) was placed close to the membrane at a distance of 5 mm from the membrane surface, while the other electrode (reference electrode) was placed 35 mm from the membrane surface. This electrode arrangement made the system sensitive to concentration changes near the membrane surface and allowed for monitoring CBL evolution over time in the chamber with low concentration. In order to ensure homogeneity of the solutions at the initial moment, a system of mechanical stirring of solutions in the chambers was used. The rotation rate of stirrers at 500 rpm ensured the minimum thicknesses of CBLs in the chambers. After filling the chambers with solutions, the solutions were mechanically stirred until the voltage between the electrodes was established (approximately 2 to 3 minutes). Turning off mechanical stirring was the beginning of CBLs reconstruction in the chambers and measurement of voltage between the electrodes. The voltage was measured with a millivoltmeter (MERATRONIK U726), with high input impedance (0.1GΩ), connected with a computer. The accuracy of the measurement was 0.1 mV. The relative difference between voltages for the cases of measurements in the same initial conditions did not exceed 5%. Voltage measurements were performed with sampling every 4s and the measurement time was 6 hours. The pressure difference between the chambers was kept at 0 Pa throughout the measurement. The membrane system was placed in a metal grounded cover ensuring the elimination of influence of external electrical fields. The measurement chamber was thermally insulated, which stabilized the temperature during the experiment. The temperature of the membrane system was stabilized at (295 ± 0.5) K. Experimental data was processed with the aid of MathCad Prime 3.0 and Origin Pro 2020 software.

## 4. Results and discussion

Configurations of the membrane system with the membrane in horizontal plane are characterized by diffusive CBLs reconstruction at the membrane surfaces in the direction of gravitational acceleration. The solutions arrangement in configuration A of the membrane system ensures the appearance of density gradients in CBLs directed as the gravitational acceleration, so that CBLs can only be reconstructed diffusively. It follows that the thickness of CBL increases over time. In this case, we can expect monotonic changes of voltage between electrodes in the membrane system [32]. On the other hand, for configuration B, there is also diffusive reconstruction of CBLs, but the increasing concentration gradients (and density gradients) in CBLs are directed opposite to the gravitational acceleration. In this case, the sufficiently large density gradients may lead to the occurrence of hydrodynamic instabilities, which are the cause of gravitational stirring of solution, not only in CBLs but also outside the CBLs. This leads to quick and random changes in concentrations at the electrode surface near the membrane, which is observed as voltage pulsations [8]. In the case of electrodes located in the chamber with lower concentration, the initial value of the observed voltage (during mechanical stirring of solutions) is equal to zero. Turning off mechanical stirring of solutions causes reconstruction of CBLs leading after some time to disturbance of concentration at the electrode located near the membrane surface and thus to a change of the measured voltage over time.

In Figs 2–5 the time characteristics of the measured voltages for configurations A (black) and B (red) are shown for gradually increasing quotients of NaCl concentrations on the membrane at the initial moment: $C_h/C_l = 100$ (Fig 2), $C_h/C_l = 500$ (Fig 3), $C_h/C_l = 1000$ (Fig 4) and $C_h/C_l = 5000$ (Fig 5). To illustrate the changes of the measured voltage at the first 50 minutes after turning off mechanical stirring of solutions, smaller plots of voltage as function of time in logarithmic scale are also presented.

The common feature of graphs 2–5 is that the initial voltage is zero. When the mechanical stirring was turned off, the voltage does not change significantly until the changes of concentrations in the chamber due to CBLs reconstruction reach the electrode located near the membrane. This is a feature common to both membrane system configurations. After a while the voltage starts to increase and as can be seen in Fig 2 there is a gradual increase in voltage for configuration A. After some time, the changes of voltage are smaller and smaller, leading to stabilization of voltage in more than 150 minutes. An increase in the initial concentration quotient on the membrane causes shortening of the time during which the rate of voltage change decreases significantly. On the other hand, for configuration B of the membrane system, after the initial time of the voltage established near zero, there is a rapid change in voltage associated with the appearance of a sufficiently large density gradients in CBLs. As a result of such density gradients there appears a convective movement of a solution with a higher density from the membrane in direction of the electrode. As a result of this convection faster changes in NaCl concentration are observed at the electrode near membrane, which causes sudden voltage change. Then the voltage pulsations appear, indicating the oscillating nature of convection in the solution near the membrane and at the surface of the active electrode. Graphs 2–5 show that the greater the concentration quotient on the membrane at the initial moment in configuration B of the membrane system the shorter the time after which a fast increase of voltage and following voltage pulsations appear. In the case of a voltage close to zero in configuration B at the beginning of CBL reconstruction, the differences in NaCl concentrations in CBL are so small that it is only the diffusion process that takes place near the membrane. On the other hand, for small initial concentration quotients on the membrane during reconstruction of CBLs over time ($C_h/C_l < 50$) in configuration B (and in configuration A) the concentration

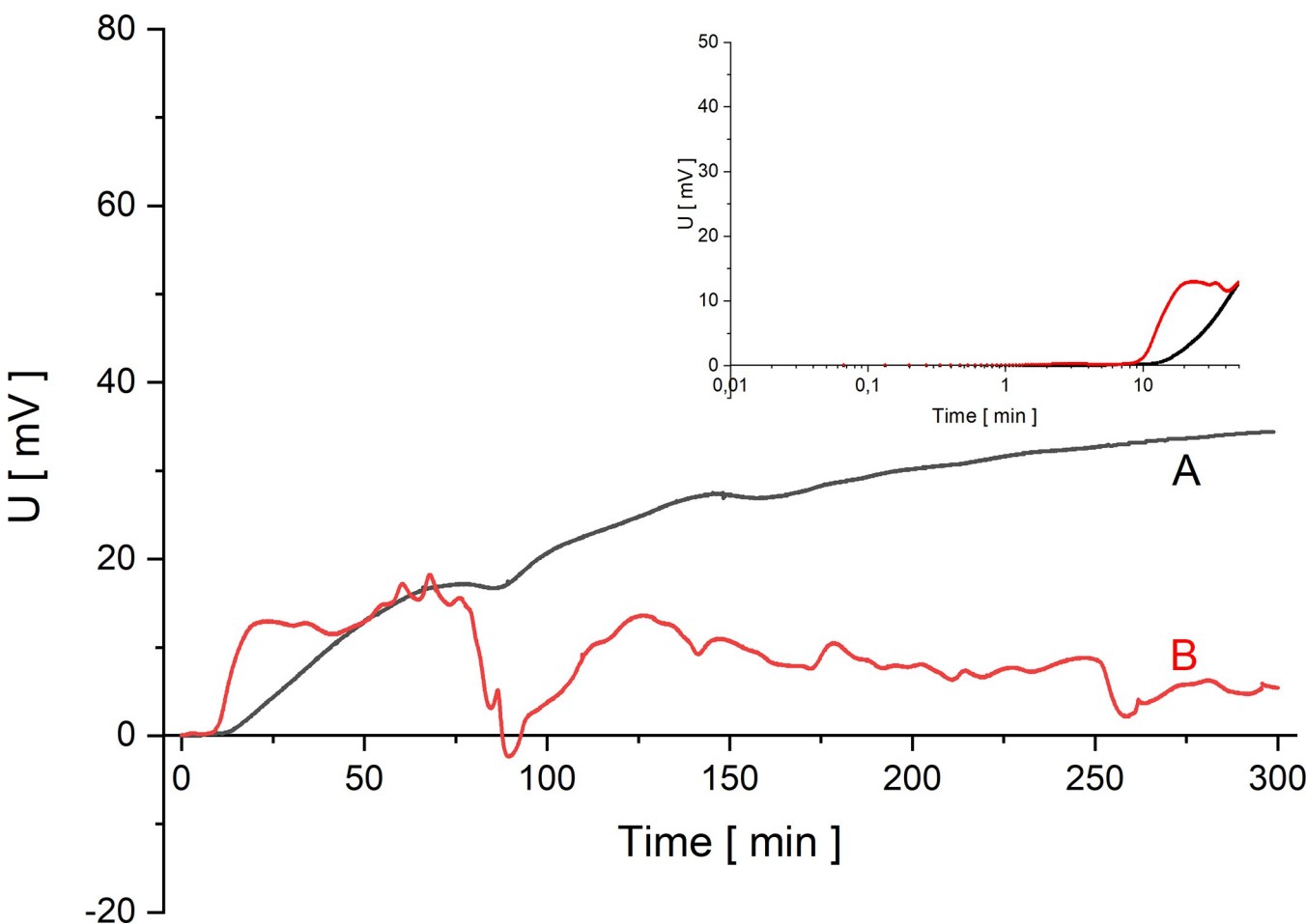

**Fig 2. Voltage between electrodes in the chamber with lower concentration as a function of time for configurations A (black) and B (red) and for $C_h/C_l = 100$.**

gradients in CBLs are so small that no hydrodynamic instabilities occur even for a long time of observation. The time characteristics of voltage in this case for both configurations A and B do not differ from each other. In Figs 4 and 5, after the common segment of time of the zero voltage, there is an increase in voltage—monotonic in configuration A and with pulsations in configuration B, connected with hydrodynamic instabilities in CBL and thus faster and random concentrations changes at the surface of the active electrode.

From the obtained curves shown in Figs 2–5, it can be seen that the initial reconstruction of the CBLs increases the initially zero voltage between the electrodes due to concentration changes at the electrode close to the membrane surface. For configuration B, in contrast to the configuration A, the time range ($T_p$) between the moment of turning off mechanical stirring and the moment of appearance of fast change of voltage with pulsations can be defined. Such changes in voltage are associated with hydrodynamic instabilities in the membrane system, arising at suitably large gradients of solution density in CBLs and opposite directed to gravitational field acceleration. Comparing the graphs for the gradually increasing $C_h/C_l$, we can observe that the time needed for the voltage pulsations to appear is shorter and shorter, and the pulsations frequency is greater and greater. As can be seen in Figs 2–5, even after 6 hours of observation the pulsations do not disappear, which demonstrates the long-term nature of hydrodynamic instabilities in the membrane system. The hydrodynamic instabilities

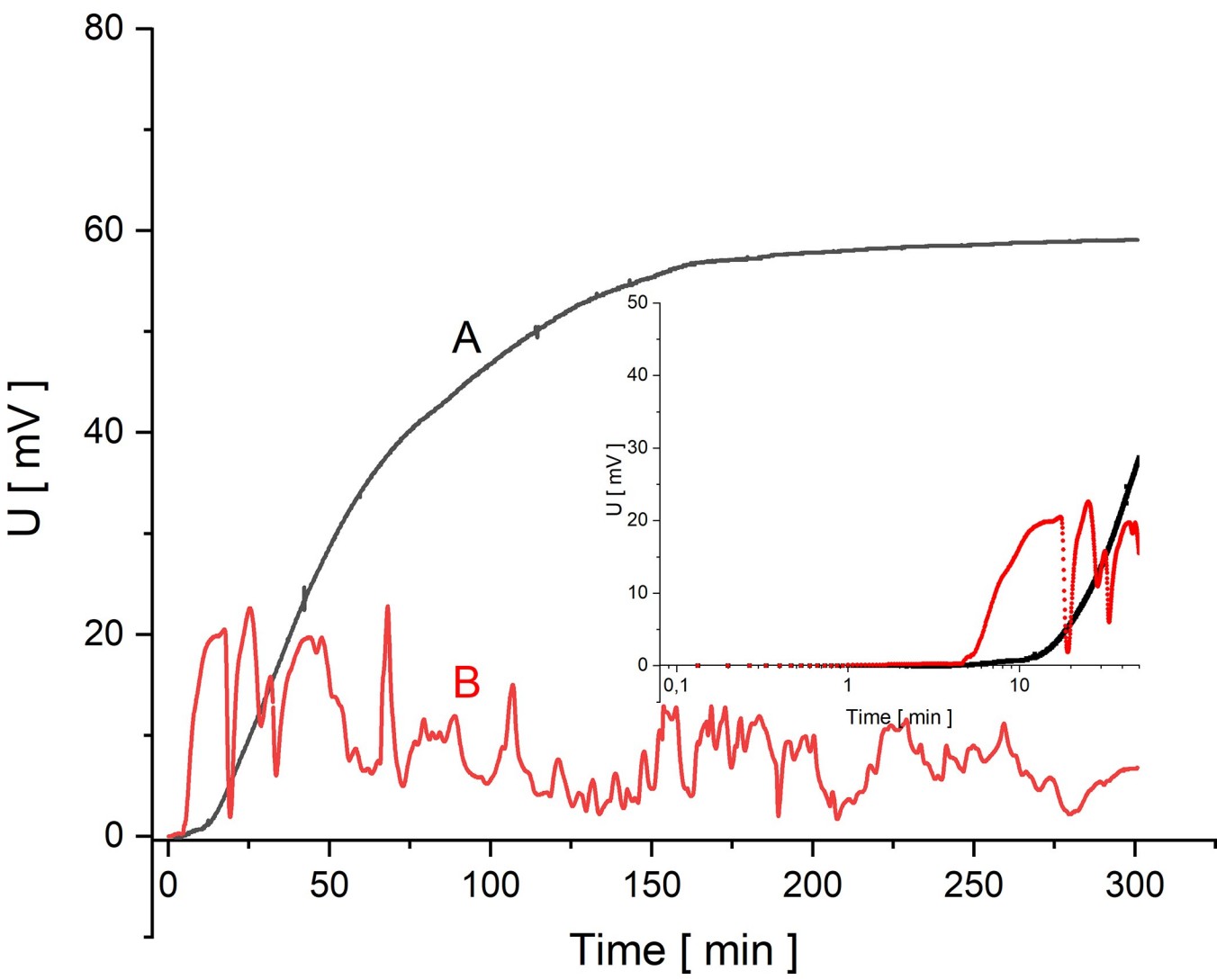

**Fig 3. Voltage between electrodes in the chamber with lower concentration as a function of time for configurations A (black) and B (red) and for $C_h/C_l = 500$.**

significantly affect the observed voltages. In the case of configuration A, the monotonic increase and then stabilization of the voltage proceeds for a sufficiently long time, while in configuration B, due to hydrodynamic instabilities, the observed voltages are significantly different from those in configuration A. Taking this into account, the steady state voltages were determined exactly 6 hours after switching off the mechanical stirring of solutions. In the case of configuration B, due to the fact that voltage pulsations were still observed after this time, the voltage in steady state was determined as averaged value in a short time, close to the 6th hour after turning off the mechanical stirring of solutions. Fig 6 shows these voltages as a function of the concentration in the chamber with higher concentration for both configurations A (black) and B (red).

As can be seen in Fig 6, for small values of the concentration quotient in the chambers (less than 50), the differences between the voltages measured in the steady states in both configurations are small. This is due to the lack of hydrodynamic instabilities during reconstruction of CBLs in configuration B. On the other hand, for $C_h/C_l$ above 50, we observe a divergence of

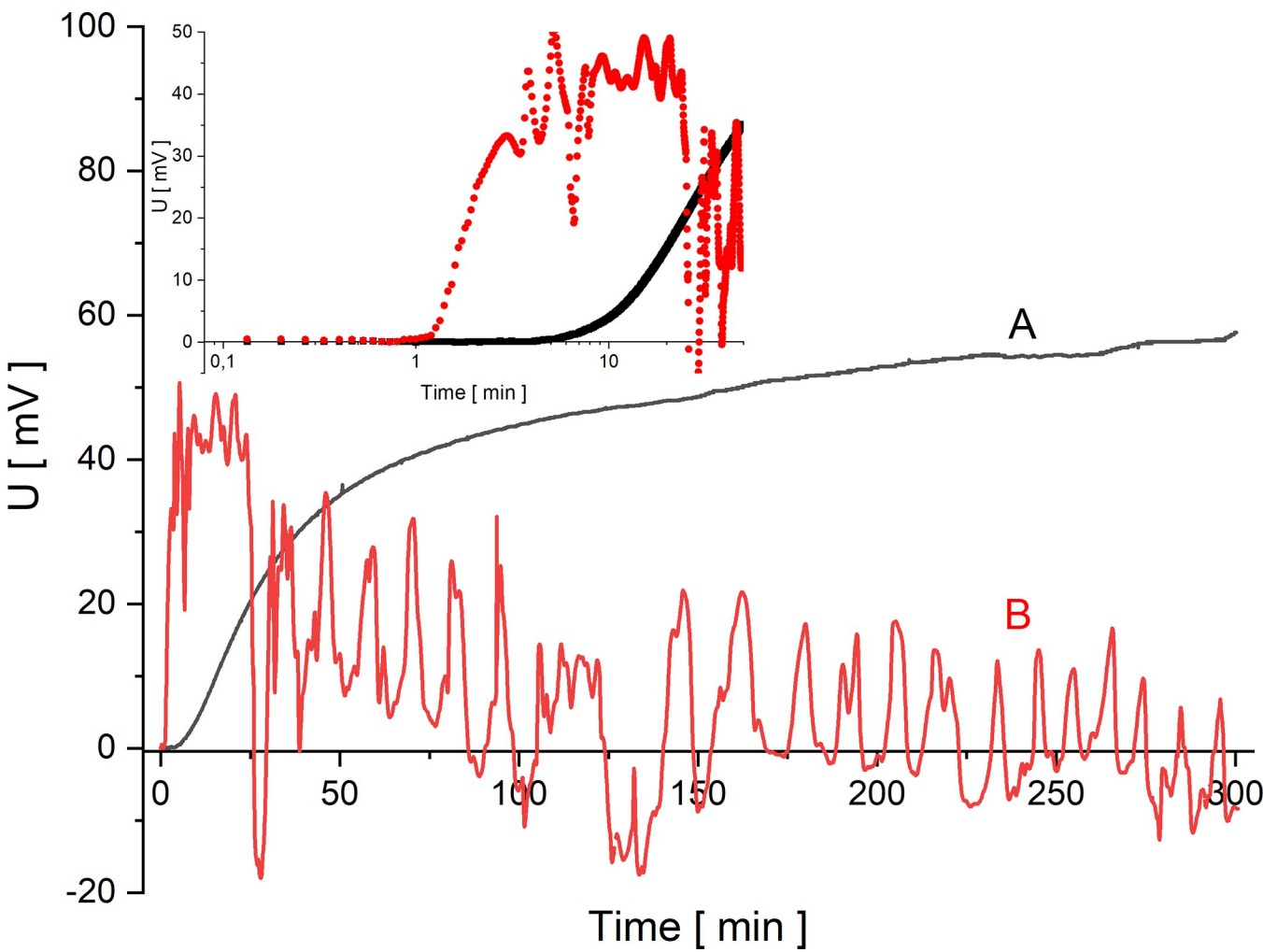

**Fig 4. Voltage between electrodes in the chamber with lower concentration as a function of time for configurations A (black) and B (red) and for $C_h/C_l = 1000$.**

the voltage concentration characteristics between the electrodes in steady states, while the increase in $C_h/C_l$ makes the differences between voltages in the steady states greater. For this reason, it can be concluded that the quotient of the initial concentrations on the membrane equal to 50 can be considered as the limit value in which the transition from diffusion to convective conditions is observed. Assuming a linear relationship between the density of the solution and its concentration, for NaCl we can write $\rho = \rho_o (1+\alpha_{NaCl} C_{NaCl})$, where $\alpha_{NaCl} = \frac{1}{\rho_o} \frac{\partial \rho}{\partial C}$ and experimentally determined $\alpha_{NaCl}$ equals to: $\alpha_{NaCl} = 0.0414 \pm 0.0003$ dm$^3$/mol. Taking into account the above-mentioned dependence it can be calculated that limit value of the quotient of the initial concentrations on the membrane ($C_h/C_l = 50$) corresponds to the initial value of the quotient of the densities of the NaCl solutions on the membrane equal to ($\rho_h/\rho_l = 1.00000229$). For configuration B and $C_h/C_l$ greater than 100 ($C_h$ greater than 0.001 mol/l), the steady state voltages are close to zero, which indicates that the intensity of gravitational stirring of solutions is high. On the other hand, the increasing value of the steady state voltage for configuration A with increasing $C_h/C_l$ is associated with greater changes in NaCl concentration at the electrode near the membrane, and with small changes in NaCl concentration at the electrode distant from the membrane.

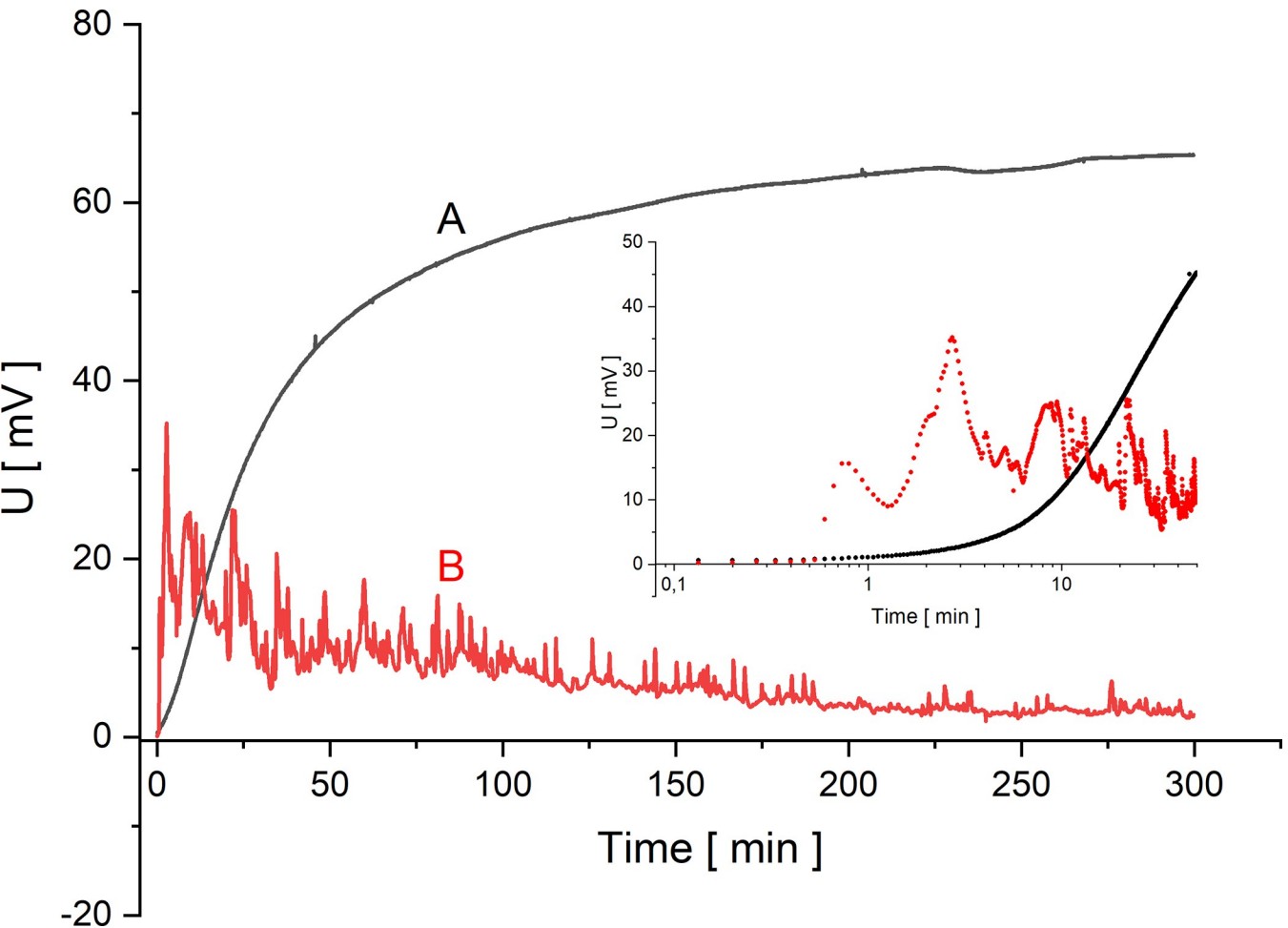

**Fig 5. Voltage between electrodes in the chamber with lower concentration as a function of time for configurations A (black) and B (red) and for $C_h/C_l = 5000$.**

The characteristic feature of the observed time characteristics of voltage for configuration B is the appearance of pulsations. The character of these pulsations depends on the initial concentration ratio on the membrane. The cause of the pulsations is the gravitational convection of solution in the chamber, caused by suitably high concentration gradient (and density gradient connected with them) in CBLs directed in the opposite direction to gravitational acceleration vector. In order to characterize pulsations the range of time from 50 min to 100 min was chosen and the number of pics was counted by means of Origin Pro 2020 software. Fig 7 shows the frequency of the observed pics (number of pics in the selected range divided by 50 min–the time range width) and amplitudes of these pics (red) as a function of initial ratio of concentrations on the membrane.

Below $C_h/C_l = 50$ the pulsations are not observed. An increase in $C_h/C_l$ causes a nonlinear increase in frequency of the observed peaks. The frequencies for all the observed $C_h/C_l$ are lower than 0.8 $min^{-1}$. The mean amplitude of peaks in the studied interval of time for small $C_h/C_l$ values is small and then increases to the maximum at $C_h/C_l$ equal to about 1100. This may be due to the fact that for sufficiently high $C_h/C_l$, (higher than 1100) the intensity of solution stirring caused by gravitational convection increases significantly. Because of the increase in frequency of the observed peaks with an increase in $C_h/C_l$, faster exchange of solution parts in

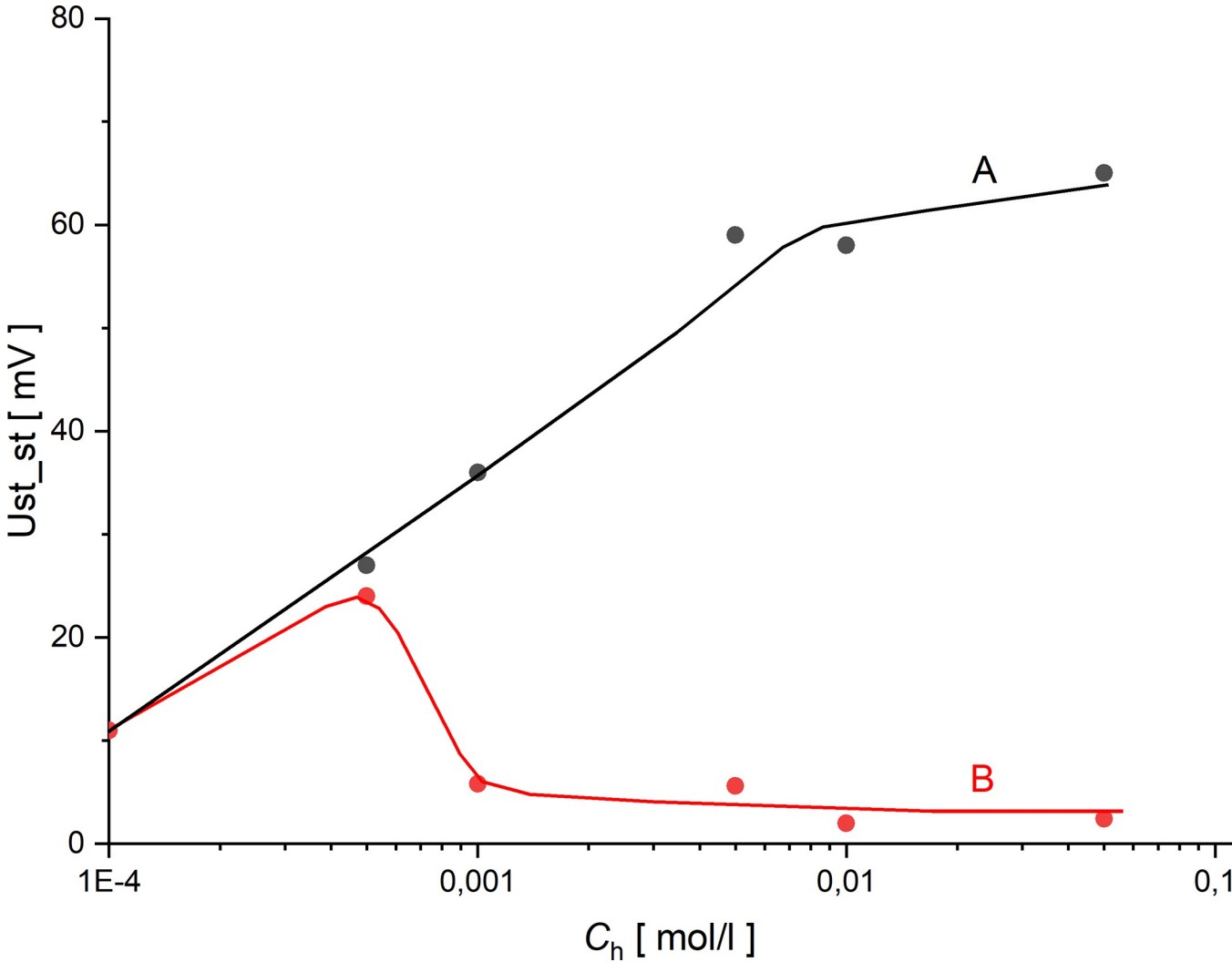

**Fig 6. Voltage in the steady state of the membrane system as a function of higher concentration for configurations A (black) and B (red).**

the adjacent areas to the membrane surfaces due to gravitational convection may proceed. Consequently, the time of peak increase substantially decreases with an increase in $C_h/C_l$, resulting in a decrease in mean amplitude of the peaks with an increase in $C_h/C_l$ above the observed maximum.

Besides, we also analysed the pulsations of voltage using the Fourier analysis of voltage signal observed in the chosen range from 50 to 100 min for different ratios of concentrations at the initial moment. The results of this analysis are presented in Fig 8A–8D. In these figures the Magnitude ($M$) is connected with real ($Re$) and imaginary ($Im$) coefficients of Fourier analysis of signal ($S$): $M(S) = \sqrt{(Re(S))^2 + (Im(S))^2}$.

As can be concluded from Fig 8A–8D the low frequencies (lover than 0.5 min$^{-1}$) dominate in all cases. However because of greater amplitudes in the intermediate $C_h/C_l$ (500 and 1000) we observe greater values of magnitudes in the whole range of frequencies. Besides, the increase in $C_h/C_l$ causes greater magnitudes for higher frequencies. From these figures we can state that the nature of voltage pulsations caused by gravitational convection is random.

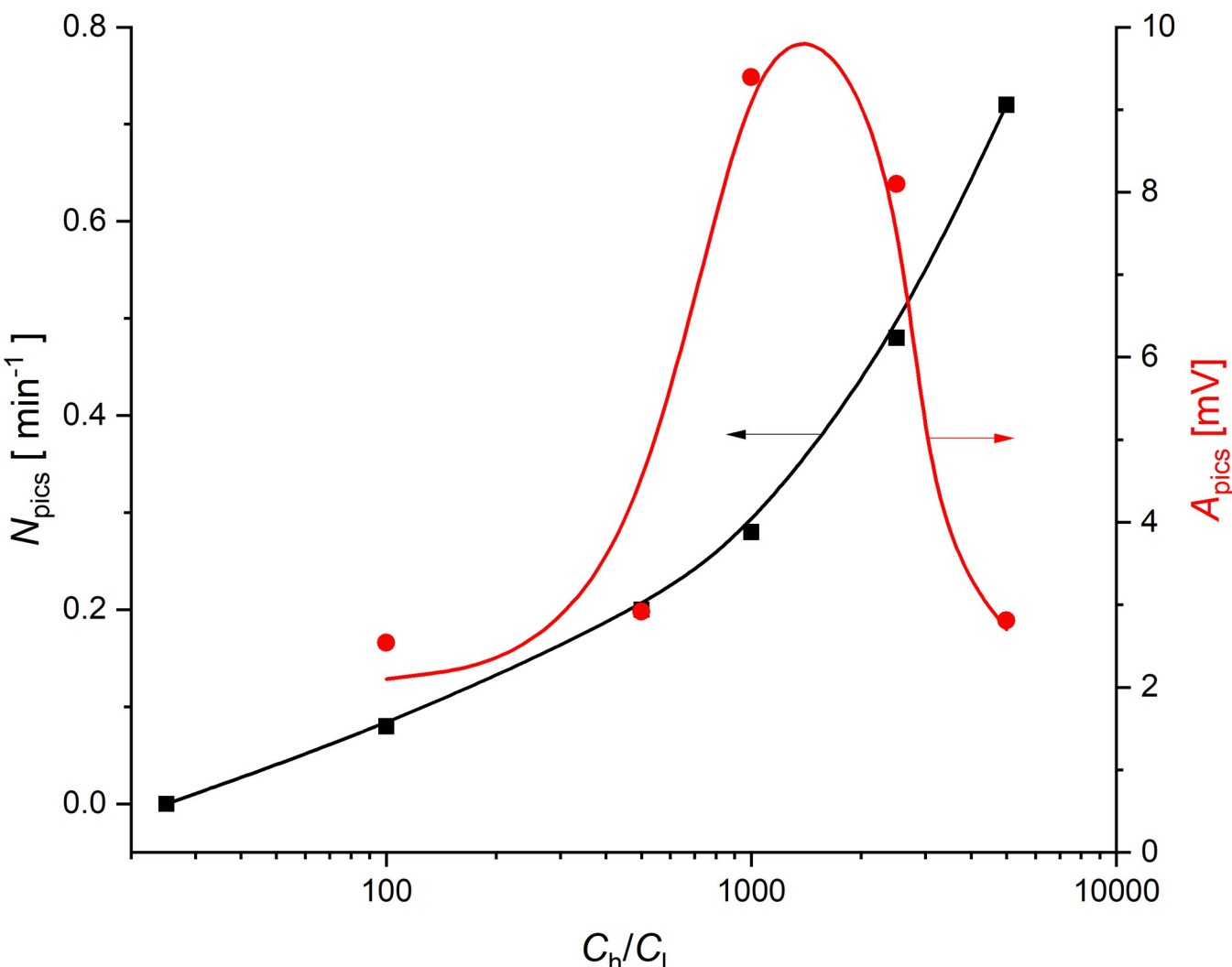

**Fig 7. Frequency of the number of peaks ($N_{pics}$) (black, left vertical axis) and average amplitude of pics ($A_{pics}$) (red, right vertical axis) observed in the range of time from 50min to 100min as a function of the quotient of the concentrations on the membrane ($C_h/C_l$) at the initial time.**

Therefore, from Fourier spectrograms we cannot select the dominant frequencies in the observed pulsations.

The CBLs reconstruction only in a diffusive manner occurs for configuration A. In the case of configuration B the diffusive reconstruction of CBLs occurs until the solution density gradients directed in opposite direction to the gravitational acceleration reach sufficiently high values. This occurs when the concentration Rayleigh number for CBL, determined by Eq (8), reaches a critical value. In this case, gravitational convection processes are activated causing effectively blur of CBLs. The elaborated model of CBLs reconstruction described by the difference Eqs (5)—(6) can be used to describe the diffusive conditions of CBLs reconstruction at the membrane surfaces. On the basis of the layers model (5)—(6), concentration ratios at the electrode surfaces for the electrodes at distances of 5 and 35 mm from the membrane were calculated. These results were compared with the course of changes of concentration quotient at the electrodes calculated on the basis of Eq (9) and voltages between the electrodes obtained from the experiment. Fig 9 shows the dependence of the concentration quotient at the

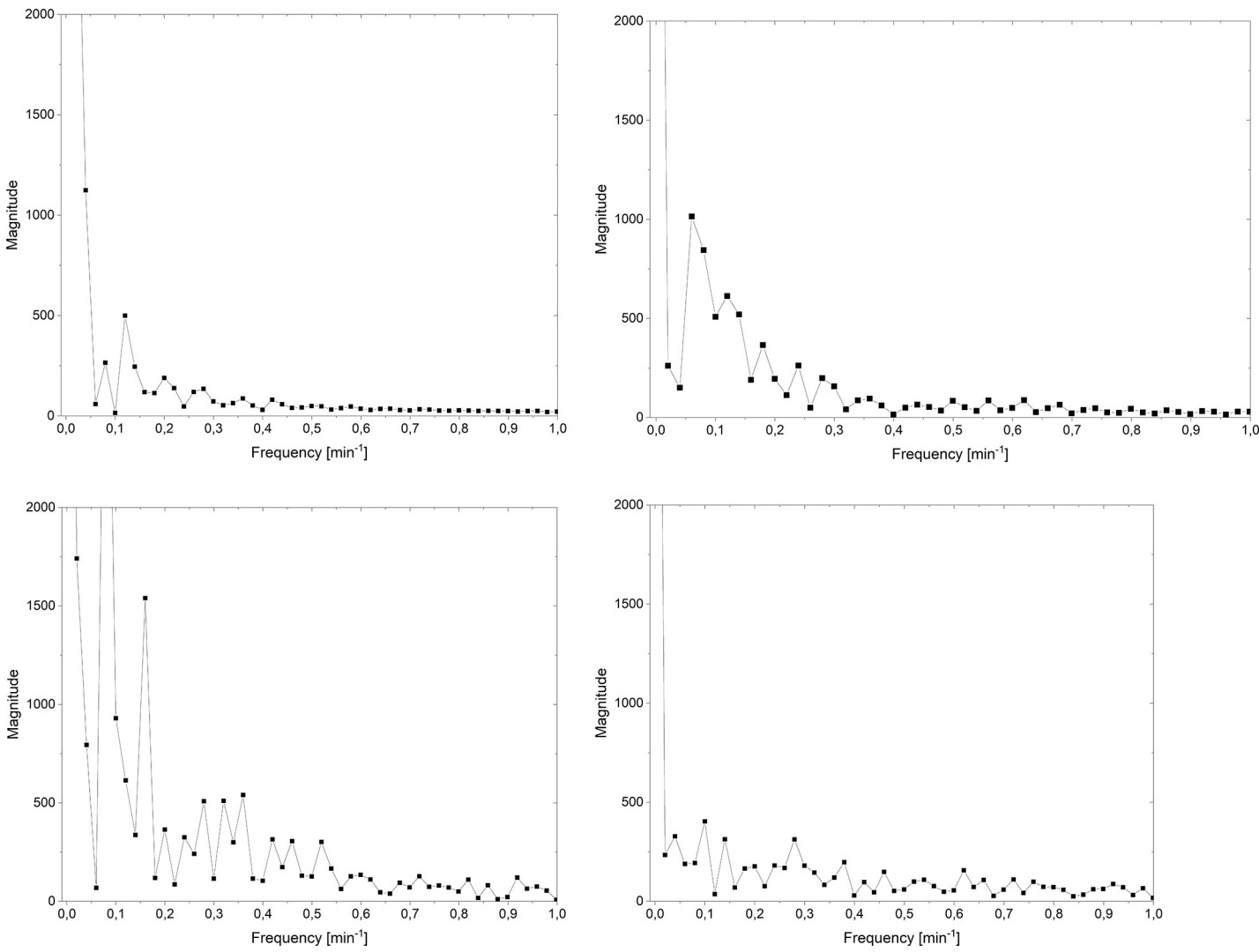

**Fig 8. Magnitude of Fourier analysis of voltage in the membrane system in a selected range of time as a function of frequency for initial conditions:** $C_h/C_l = 100$ (a), $C_h/C_l = 500$ (b), $C_h/C_l = 1000$ (c), $C_h/C_l = 5000$ (d).

electrode surfaces as a function of time for the initial concentration quotients on the membrane: 100, 500 and 1000, respectively, calculated from the model (5)—(6) (solid lines) and data from the experiment (points). In the case of experimental data (points in Fig 9), the measured voltages between the electrodes in the first 50 minutes of CBL reconstruction in configuration A were transformed on the basis of Eq (9) into quotients of NaCl concentrations at the electrode 5 mm from the membrane ($C_{elm}$) and at the electrode 35 mm from the membrane ($C_{elk}$).

As can be seen in Fig 9 in the initial range of time after switching off the mechanical stirring of solutions, i.e. in the diffusive reconstruction of CBLs, the concentration ratio at electrode surfaces ($C_{elm}/C_{elk}$) is approximately constant and equal to 1 (the voltage between the electrodes in that range of time is approximately equal to zero). This is connected with the fact that in order for the voltage between the electrodes to begin to change, the CBL must be reconstructed enough to cause a change of concentration at the electrode close to the membrane surface. Further reconstruction of CBL causes changes of concentration at the surface of the

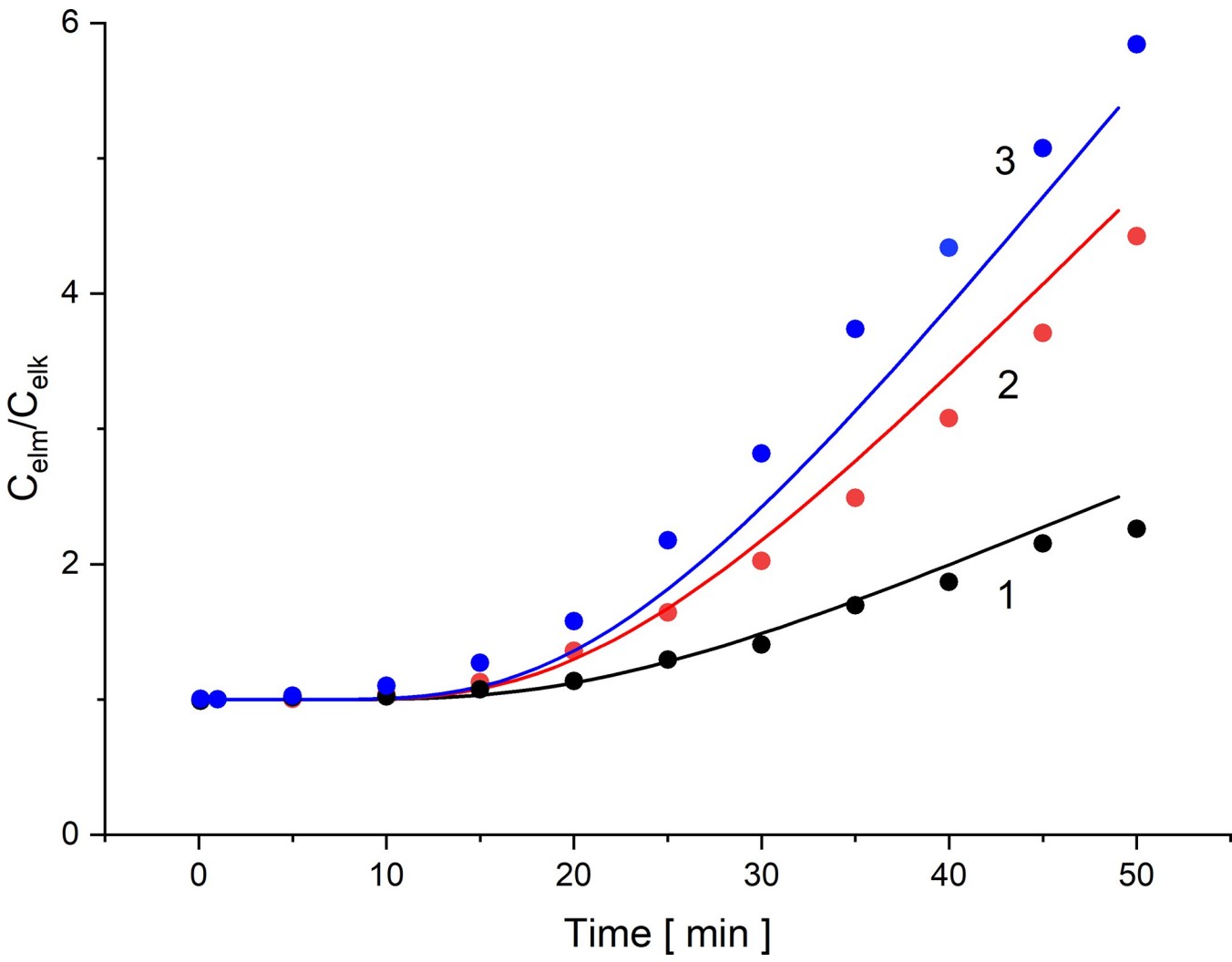

**Fig 9. Concentration quotient at the electrode surfaces as a function of time after switching off the mechanical stirring of solutions, for the initial concentration quotients on the membrane ($C_h/C_l$): 100 ($C_h = 1$ mol/m$^3$) (1), 500 ($C_h = 5$ mol/m$^3$) (2) and 1000 ($C_h = 10$ mol/m$^3$) (3).**

electrode closer to the membrane, leading to a gradual increase in voltage between the electrodes due to the increasing difference between concentrations at electrode surfaces in the chamber with a lower NaCl concentration. The differences obtained between the data from the model and the experiment result from the approximations to the geometry of the system assumed in the model of CBLs reconstruction at the membrane surfaces.

On the other hand, only diffusive reconstruction of CBLs in configuration B occurs until hydrodynamic instabilities appear. The beginning of the appearance of hydrodynamic instabilities can be determined on the basis of the Rayleigh concentration number described by Eq (8). In this case for calculation of concentration Rayleigh number the thickness of CBLs determined by the Lerche criterion (7) is needed. Therefore, using Eqs (5)—(9) we calculated the concentration Rayleigh numbers for CBL in the chamber with a lower concentration as functions of time after switching off the mechanical stirring of solutions, which is presented in Fig 10 for $C_h/C_l = 100$, 500, and 1000.

The value $(Ra)_C = 1100{,}6$ [29] was taken as a critical value of the concentration Rayleigh number. The time needed for the concentration Rayleigh number to reach $(Ra)_C$ is defined as

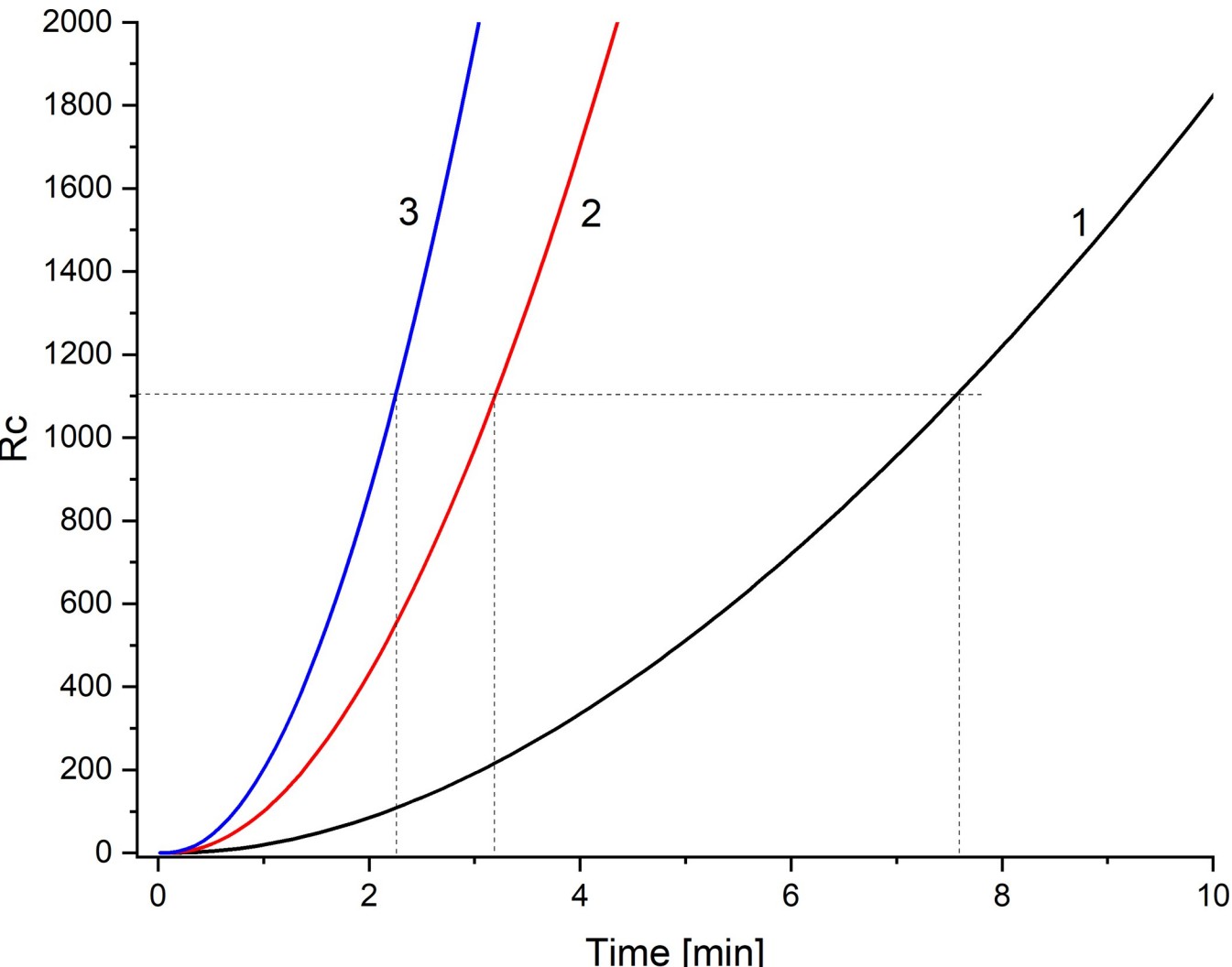

**Fig 10. Concentration Rayleigh number as a function of CBL reconstruction time, for initial concentration quotients on the membrane $C_h/C_l$ = 100 (1), 500 (2) and 1000 (3).**

$T_p$. This is a time needed for hydrodynamic instabilities to appear in the membrane system, leading to convective stirring of solutions in the chambers. Using the model (5)—(6) we determined the CBL thickness criterion (7) and the concentration Rayleigh number defined by Eq (8), $T_p$ for different values of $C_h/C_l$, which is shown in Fig 11 as a solid line. $T_p$ was determined from the experiment as the beginning of a fast change and then voltage pulsation in configuration B of the membrane system. The experimentally determined $T_p$ ($T_p^{-1}$ as function of $C_h/C_l$) for configuration B of the membrane system is shown in Fig 11 as points.

As can be seen in Fig 11, an increase in initial concentration quotient on the membrane in configuration B results in increasing reciprocal of $T_p$ (decrease of $T_p$). This dependence is non-linear. For the initial concentration quotient on the membrane lower than 50, hydrodynamic instabilities were not observed in the membrane system (no voltage pulsation) so $T_p^{-1} = 0$.

The research results presented in the paper show that the membrane system with electrodes placed in a solution with lower concentration can be used as a gravitational field detector. These results may be helpful in explaining the mechanisms of adaptation to the changing

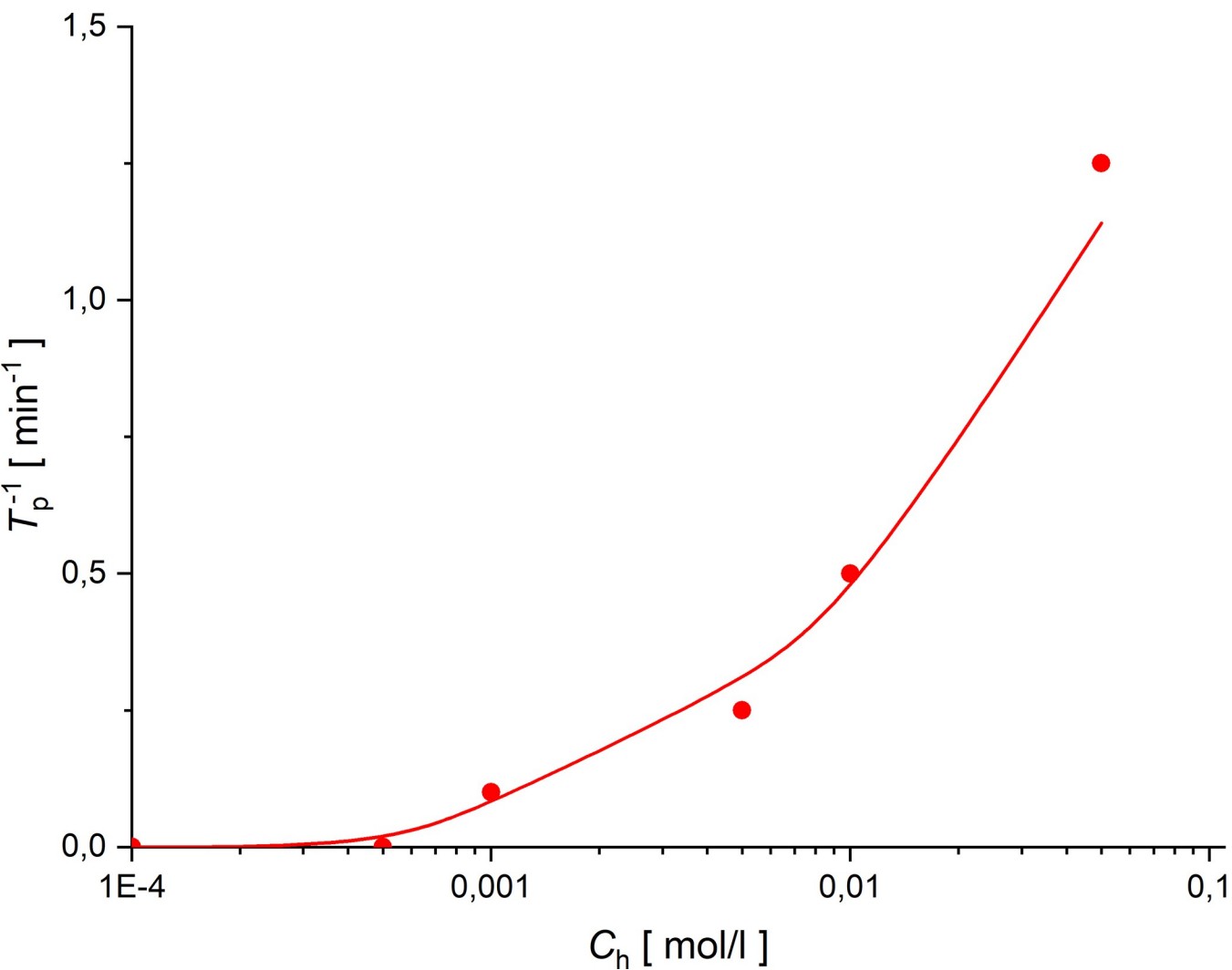

**Fig 11. Inverse of time of appearance of hydrodynamic instability as a function of initial concentration $C_h$ ($C_l = 10^{-5}$ mol/l) for configuration B of the membrane system.**

gravitational field in living organisms [33–36]. Solute flows through the membrane, as well as diffusive and convective flows in the vicinity of the membrane take place in a homogeneous gravitational field. For this reason, they can be included into the basic group of transport phenomena occurring at all levels of organization of both natural and artificial systems. The driving stimuli of these transports are a manifestation of the existence of various physical fields involved in shaping the field character of nature [37]. Important stimuli in the problems discussed in this article are density gradients related to the density and concentration fields in the nearest vicinity of the membrane, or gradients of concentrations and electrical potentials on the membrane connected with the transport of electrolytes through the membrane. The result of the appearance of these fields is the concentration polarization of the membrane, the difference of electric potentials observed in the membrane system and gravitational convection in the areas adjacent to the membrane. The concentration polarization of the membrane, which is the result of the diffusive (molecular) creation of CBLs in the membrane surroundings, reduces membrane transport. It occurs both in biological and artificial systems. The reason for the reduction of membrane transport is the decrease in concentration gradients of transported

substances on the membrane. In artificial systems, under the conditions of relationship between the density gradient and the gravitational field, the reduced concentration gradients on the membrane may be partially restored by gravitational instabilities such as gravitational convection in the vicinity of the membrane. Another way to reduce the concentration polarization of the membrane is to obtain homogeneous solutions in the chambers of the membrane system by intensive stirring of the solutions. In this case, due to the elimination (in fact, minimization) of the formation of CBLs, the effects associated with the gravitational field disappear. However, this requires the work of external forces, significantly disturbing the processes taking place in the membrane system without the participation of external factors. In biological systems, similar processes could be found in the work performed by molecular membrane pumps, keeping the stimuli on the membrane at a fixed, stationary levels. The appearance of the membrane concentration polarization and its reduction (gravitational, mechanical or chemical) may in some cases be a natural regulator of transports through the membrane and thus also the rate of entropy production [33]. This type of phenomenon could be advantageous for the cell to slow down the aging process of the cells and the whole organism.

## 5. Conclusions

- Localization of the electrodes in the chamber with a lower concentration at different distances from the membrane allows for more accurate observations of the dynamic phenomena of CBLs reconstruction and their destruction due to the appearance of hydrodynamic instabilities.

- The results obtained for NaCl solutions and other electrolytes (KCL) are similar. There are analogous characteristics of changes in temporal and concentration voltages in the membrane system connected with the diffusion and convection conditions of CBLs reconstruction in the system with a membrane in the horizontal plane.

- The initial voltages between the electrodes localized in one of the chambers are zero, and over time, non-zero voltages arise depending on the CBL state in that chamber.

- The higher the concentration of solution in the chamber with electrodes the lower the sensitivity of the system to illustrate the dynamic changes in the concentration of solutions in the CBL area.

- For appropriately large values of the initial concentration quotient on the membrane (for NaCl aqueous solutions: $C_h/C_l > 50$), the time characteristics of the voltages of configurations A and B of the membrane system differ significantly due to the possibility of appearance of hydrodynamic instabilities in configuration B.

- The monotonic increase of voltage over time for configuration A is related only to diffusive CBLs reconstruction.

- Pulsations of voltage in configuration B are connected with the appearance of hydrodynamic instabilities in the membrane system, while for larger initial concentration ratios on the membrane, the intensity of pulsations (their frequency) is higher.

- The time needed for the appearance of hydrodynamic instabilities in the membrane system depends on $C_h/C_l$. An increase in $C_h/C_l$ makes this time shorter.

- Concentration Rayleigh number for CBL as a function of time of CBLs reconstruction for the case of both electrodes in one chamber depends on $C_h/C_l$. An increase in $C_h/C_l$ causes greater growth dynamics in time of $R_C$ and thus a shorter time is needed to reach the critical $R_C$ value.

## Supporting information

**S1 Data.**

(XLSX)

## Author Contributions

**Conceptualization:** Sławomir Grzegorczyn, Andrzej Ślęzak.

**Data curation:** Sławomir Grzegorczyn.

**Formal analysis:** Sławomir Grzegorczyn.

**Funding acquisition:** Sławomir Grzegorczyn.

**Investigation:** Sławomir Grzegorczyn.

**Methodology:** Sławomir Grzegorczyn, Andrzej Ślęzak.

**Project administration:** Sławomir Grzegorczyn.

**Validation:** Sławomir Grzegorczyn, Andrzej Ślęzak.

**Visualization:** Sławomir Grzegorczyn.

**Writing – original draft:** Sławomir Grzegorczyn.

**Writing – review & editing:** Sławomir Grzegorczyn, Andrzej Ślęzak.

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
