## [Decision Letter · Decision Letter 0]

18 Sep 2021

PONE-D-21-16987Study of thin layer film evolution near  bacterial cellulose membrane by Ag|AgCl electrodes in chamber with lower concentrationPLOS ONE

Dear Dr. Grzegorczyn,

Thank you for submitting your manuscript to PLOS ONE. After careful consideration, we feel that it has merit but does not fully meet PLOS ONE’s publication criteria as it currently stands. Therefore, we invite you to submit a revised version of the manuscript that addresses the points raised during the review process. Reviewer's comments are appended below. 

We look forward to receiving your revised manuscript.

Kind regards,

Mahendra Singh Dhaka, Ph.D.

Academic Editor

PLOS ONE

Journal Requirements:

Reviewers' comments:

Reviewer's Responses to Questions

**Comments to the Author**

1. Is the manuscript technically sound, and do the data support the conclusions?

Reviewer #1: Yes

Reviewer #2: Yes

2. Has the statistical analysis been performed appropriately and rigorously? 

Reviewer #1: Yes

Reviewer #2: Yes

3. Have the authors made all data underlying the findings in their manuscript fully available?

Reviewer #1: Yes

Reviewer #2: Yes

4. Is the manuscript presented in an intelligible fashion and written in standard English?

Reviewer #1: Yes

Reviewer #2: No

5. Review Comments to the Author

Reviewer #1: 1. Provide photos of reaction setup taken during analysis.

2. Morphology of the utilized membrane is missing; you can use any of mentioned techniques AFM, SEM or TEM and add its analysis in your manuscript.

3. Why only NaCl is used as electrolyte whether it will work on KCl, or any uni-uni electrolyte, how it will affect the result if uni-bi or bi-bi electrolytes were taken in account, explain your answer in conclusion.

4. Which bacterial membrane was utilized is not mentioned; also add extraction and preparation process of stable film.

5. XRD, FTIR is required.

6. How does temperature affect the result?

7. Add antimicrobial activity of the film used so as its application can be enhanced.

8. There are some typographical and grammatical mistakes.

Reviewer #2: The authors have written very well the manuscript with scientific interpretations. Manuscript deals with the Study of thin layer film evolution near bacterial cellulose membrane by Ag|AgCl electrodes in chamber with lower concentration. Following points should be considered for further improvement of the presented work:

In artificial systems, under the conditions of relationship between the density gradient and the gravitational field, give the region the reduction in concentration gradients on the membrane.

There are few grammatical mistakes. Please improve and reduce it for improvement the manuscript.

A divergence of the voltage concentration characteristics between the electrodes in steady states is observed, while the increase in Ch/Cl. Why? What is novelty of the present work?

6. PLOS authors have the option to publish the peer review history of their article (what does this mean?). If published, this will include your full peer review and any attached files.

Reviewer #1: No

Reviewer #2: No

---

## [Author Response · Author response to Decision Letter 0]

24 Oct 2021

Responses to reviewers' comments are included in the attached file <<Response to Reviewers>>

---

## [Decision Letter · Decision Letter 1]

12 Jan 2022

Study of thin layer film evolution near  bacterial cellulose membrane by Ag|AgCl electrodes in chamber with lower concentration

PONE-D-21-16987R1

Dear Dr. Grzegorczyn,

We’re pleased to inform you that your manuscript has been judged scientifically suitable for publication and will be formally accepted for publication once it meets all outstanding technical requirements.

Kind regards,

Mahendra Singh Dhaka, Ph.D.

Academic Editor

PLOS ONE
---

## [Editor Report · Acceptance letter]

24 Jan 2022

PONE-D-21-16987R1 

Study of thin layer film evolution near  bacterial cellulose membrane by Ag|AgCl electrodes in chamber with lower concentration 

Dear Dr. Grzegorczyn:

I'm pleased to inform you that your manuscript has been deemed suitable for publication in PLOS ONE. Congratulations! Your manuscript is now with our production department. 

Kind regards, 

on behalf of

Dr. Mahendra Singh Dhaka 

Academic Editor

PLOS ONE